# Diet, Food Intake, and Exercise Mixed Interventions (DEMI) in the Enhancement of Wellbeing among Community-Dwelling Older Adults in Japan: Systematic Review and Meta-Analysis of Randomized Controlled Trials

**DOI:** 10.3390/geriatrics9020032

**Published:** 2024-03-04

**Authors:** Takaaki Miyazaki, Toshihiro Futohashi, Hiroki Baba

**Affiliations:** 1Department of Rehabilitation, Tokyo University of Technology, 5-23-22 Nishikamata Ota-Ku, Tokyo 144-8535, Japan; futohashith@stf.teu.ac.jp; 2Department of Rehabilitation, Heisei Yokohama Hospital, 550 Totsukamach Totsuka-Ku, Yokohama 244-0003, Japan; baba.hiroki@hmw.gr.jp

**Keywords:** diet, food, exercise, old, Japan

## Abstract

This systematic review and meta-analysis discusses the available data on the efficacy of diet, food intake, and exercise mixed interventions (DEMI) for community-dwelling older adults in Japan and assesses the evidence level. We searched the literature regarding the research questions using electronic and hand-searching methods. To ensure the reliability and quality of the evidence, we used the Cochrane risk of bias tool and GRADE system. All studies included DEMI; other interventions included group activities, health education, and community participation. All interventions were categorized into three classifications, namely “Diet and food intake”, “Exercise”, and “Other”. Programs included lectures, practical exercises, group activities, consulting, and programs that could be implemented at home. By comparing groups and measuring outcomes at various time points, most studies reported positive results regarding the impact of the interventions. Specifically, usual gait speed, Food Frequency Questionnaire Score, and Diet Variety Score demonstrated significant improvement. Additionally, three studies demonstrated improvement in frailty. This review suggests that DEMI resulted in improvements in some outcome variables. However, the efficacy of all variables was not fully examined. The results of the meta-analysis revealed positive outcomes for some variables, although the evidence level for these outcomes was considered moderate.

## 1. Introduction

In an aging society, older adults face various challenges related to their physical functioning in daily life, as well as issues concerning nutrition. To address these concerns, a mixed intervention involving nutrition and exercise has been implemented to improve conditions such as sarcopenia, frailty, and the aging process in older individuals [1,2]. Numerous studies have demonstrated the efficacy of such interventions for older adults [3].

Japan garners significant attention for having one of the highest levels of long-term healthy life expectancy and overall life expectancy at birth worldwide [4]. Distinctive initiatives in Japan, such as the “Smart Life Project of Japan”, ref. [5] affirm that the older population in Japan enjoys exceptional health and functional status due to health and care services that emphasize exercise and dietary practices. We believe that researching nutritional intake and exercise interventions for community-dwelling older adults in Japan is extremely important for the future of their health.

The innovative long-term care insurance system (LTC) in Japan provides comprehensive coverage for older adults. The health status of these individuals is influenced by both health and care services and the compensation system within the social and healthcare framework, particularly in terms of nutritional intake. 

According to many studies previously reported in Japan [6,7], significant concerns regarding nutrition and exercise mixed interventions exist. A prior systematic review highlighted the low level of evidence available and the necessity for future research [8]. Earlier studies have reported promising results for community-dwelling older individuals. However, the number of eligible, well-designed studies has been limited. Most prior studies had poor study designs, a small number of participants, or were lacking a comparison group. Furthermore, these studies often lacked comprehensive details about the interventions employed.

In the daily lives of the older adults, it is crucial to consider their habitual activities, especially those related to daily eating. When implementing interventions for this population, it is advisable to address not only nutritional aspects but also the broader context of eating, food, and other relevant psychosocial factors. Despite the importance of dietary and exercise factors for community-dwelling older adults, many studies [6,7,9,10,11,12,13,14] have not emphasized the significance. However, there is a noticeable gap when it comes to investigating the efficacy of interventions that combine both food and exercise for the older adults. This gap is largely due to the lack of well-designed studies; lack of comprehensive information about the participants, their characteristics, and dietary and food interventions; and missing details of the specific interventions themselves.

In Japan, the policy task force overseeing care programs [15,16] has shown a clear commitment to improving nutrition. However, studies addressing the impact of this policy on nutritional improvement have been insufficiently examined and discussed [8]. One study with a substantial sample size investigated both formal and informal programs that involved exercise and nutrition interventions for community-dwelling older adults [16]. Especially, the LTC of Japan offers care services for individuals with various degrees of functional disabilities, including food and exercise mixed interventions [15,16]. These services are accessible to older individuals upon certification of their needs. However, despite the availability of these services, there remains a lack of sufficient evidence regarding the efficacy of these interventions. While a few studies, such as the one conducted in Kameoka on LTC in Japan, attempted to verify their efficacy, the authors lacked comprehensive evaluations, control groups, and the revelation of individual factors or detailed issues [16].

We hypothesize that DEMI is effective for community-dwelling older adults in Japan. This review aims to discuss the efficacy of diet, food intake, and exercise mixed intervention (DEMI) for community-dwelling older adults in Japan and assesses the supporting level of evidence for these interventions. To this end, this systematic review and meta-analysis was conducted to investigate the efficacy of DEMI for community-dwelling older adults in Japan and determine the level of supporting evidence. We scoped and systematically reviewed previous articles related to this thesis, exploring the efficacy, specific approaches, and future research directions.

## 2. Materials and Methods

### 2.1. Protocol of Systematic Review

We implemented this systematic review following standard procedures [17] and in accordance with the Preferred Reporting Items for Systematic reviews and Meta-Analyses (PRISMA) guidelines [18]. Our systematic search and the selection process for eligible studies are outlined in Figure 1, as detailed in our protocol. 

CINAHL: Cumulative Index to Nursing and Allied Health Literature.

### 2.2. Review Questions

The review question addressed in this study was “What is the efficacy and methodology of DEMI for community-dwelling older adults in Japan?”

### 2.3. Aim of This Review

The aim of this review is to clarify issues concerning the efficacy of DEMI for the older adults in Japan as follows: the level of evidence provided by the previous studies, specific programs and interventions implemented, and the efficacy of these interventions.

### 2.4. Research Strategy

The literature search, encompassing previously published research relevant to our study questions, was conducted through a combination of electronic and manual searching, led by a single author (T.M.). After excluding duplicate and noneligible publications, the remaining literature was evaluated and critically reviewed by all authors (T.M., H.B., and H.F.). An assessment of the quality and certainty of the included studies was performed by all authors (T.M., H.B., and H.F.). The meta-analysis was done by one author (T.M.), and the results of both the systematic review and meta-analysis were discussed and reviewed by all authors (T.M., H.B., and H.F.).

### 2.5. Searching Methods

The literature search involved the use of several electronic databases, including Medical online [19], Ichu-shi web [20], PubMed [21], Cumulative Index to Nursing and Allied Health Literature (CINAHL) [22], and Cochrane [23]. In addition, a manual search was conducted from selected literature sources. A combination of Medical Subject Headings (MeSH) terms and keywords, in both English and Japanese, were used for the search either singularly or in combination. In English, the terms included “old”, “elderly people”, “nutrition”, “diet”, “food”, “eating”, “physical”, “exercise”, “function”, “muscle”, “mobility”, “activity”, “gait”, “walking”, “balance”, “community”, “district”, “dwelling”, “living”, “house”, “resident”. In Japanese, synonymous keywords were “Nippon”, “Koureisha”, “Chiiki-Zaizyu”, “Eiyou”, “Syokumotsu”, “Shoku-Ji”, “Tairyoku”, “Undou”, “Tiiki”, and “Zaitaku”.

### 2.6. Protocol for Selecting Eligible Studies

We initiated our search by querying a database. In the first step, one author, T. F., conducted an initial selection of literature matching the specified keywords and excluded any duplicate abstracts. Next, the inclusion and exclusion criteria were applied to the selected abstracts by all authors (T.M., H.B., and H.F.). The process of moving from the initial pool of studies to the final selection of studies was facilitated using the literature management software, Rayyan (PC ver.) [24], which enables seamless collaboration among multiple reviewers. This matching and discussion phase continued until a consensus was reached among the authors. In the third step, all authors collectively assessed the selected literature based on their abstracts. Finally, the authors examined the full texts of all studies, and the selection of eligible studies was made after a critical review conducted by each of the three reviewers.

### 2.7. Inclusion and Exclusion Criteria

All authors selected literature that met the eligibility criteria based on the following inclusion and exclusion criteria for the type of study, participants, and interventions.

The review on the efficacy of DEMI for community-dwelling older adults in Japan included the following criteria:

Inclusion criteria were as follows: (1) full-text literature; (2) in IMRAD format (Introduction, Methods, Results, and Discussion); (3) randomized controlled trial (RCT) study design; (4) case-comparison group comparison studies between intervention and non-intervention or placebo-control groups; (5) with English or Japanese as the language of publication; (6) study participants aged 65 years and older; (7) residing in community dwellings in Japan; (8) receipt of combined diet, food intake, and exercise interventions; and (9) published between April 2000 and January 2021, since LTC for older adults in Japan was started on 1 April 2000.

In this review, the intervention programs referred to as “DEMI” encompass a combination of interventions targeting eating behavior and exercise among community-dwelling older individuals in Japan. Eating behavior, in this context, means not only food and meals but also the entire process including meal preparation, cooking, eating, and meal cleanup, and includes instruction, consultations, and group-based practical activities related to eating behavior. Additionally, exercise interventions are provided, which can be either self-guided or instructor-led. The manner of intervention for exercise is limited to face-to-face interactions, such as personal or group exercise sessions. This excludes indirect intervention methods conducted at home, such as using telephone support, written communication (letters), electronic devices, or other remote communication tools.

Exclusion criteria were as follows: (1) case reports, qualitative studies, government or institution reports, conference reports, doctoral theses, or book chapters; (2) with severe disability impeding participant’s independence in any daily activities; (3) studies without a combination of interventions, or those without any form of intervention; and (4) studies involving one group analysis without a comparison control or placebo group.

### 2.8. Data Extraction

One author (T.M.) was responsible for data extraction from the eligible studies and subsequently presented the results in a single table. This table includes key information such as the authors of the study, publication year, study design, participant details, descriptions of interventions, and the reported outcomes.

### 2.9. Assessment of Quality and Risk of Bias of the Included Studies

The quality assessment of the included studies was conducted by all three authors (T.M., T.F., and H.B.) using the “Risk of Bias ver 2” (RoB 2) [25,26] tool developed by Cochrane. The RoB 2 is recognized for its reliability and validity in assessing the quality of studies. It evaluates six key components, namely the randomization process, deviations from intended interventions, missing outcome data, measurement of the outcome, selection of the reported result, and overall bias. The ratings for each of these components were categorized into three grades, namely low risk of bias, unclear risk of bias, and high risk of bias. In instances where there were discrepancies in the risk of bias assessments among the three reviewers, these discrepancies were resolved through discussion and consensus until an agreement was reached.

### 2.10. Primary and Secondary Outcomes

We established that primary outcomes were directly affected, while the secondary outcomes were influenced by the changes in the primary outcomes or through synergistic effects. The primary outcomes primarily focused on nutrition and physical elements. However, considering the contents and mechanism of DEMI, we further assumed that DEMI could also impact secondary outcomes, which included functional mobilities, activities, and psychosocial changes. 

All outcomes were categorized into the following groups: “frailty”, “physical functions” (Phy F), “psychosocial functions” (Psy F), “nutritional status” (Nu), “food intake” (F), “behavior (frequency, duration)” (B), and “other outcome factors” (O).

### 2.11. Effect Measures

To assess the significant effects of DEMI interventions, we compared the results of outcome variables in the reviewed studies and identified significant changes. The effects of DEMI were measured by comparing the mean differences (MDs) of outcomes between pre- and post-interventions using meta-analysis if two or more comparable variable values were available. The results were visualized using forest plots. A random-effects model was utilized, and heterogeneity was provided with I^2^. 

### 2.12. Synthesis Methods

In the eligible studies that satisfied the inclusion criteria, we employed meta-analysis as the synthesis method for outcomes. A meta-analysis was conducted when pre- and post-intervention data were extractable. In instances where the compared data were not explicitly reported, they were tabulated, and a qualitative descriptive analysis was conducted as part of the systematic review. Only ordinary scale data and interval scale data were extracted for analysis, excluding nominal data.

Regarding statistical methods, the number of eligible studies was determined using the PRISMA checklist [27]. To structure our analysis, we followed the “patient/population, intervention, comparison and outcomes” (PICO) framework, which helped to systematically examine and discuss the key characteristics and outcomes of the included studies. Statistical analysis was performed using the statistical software EZR [28]. The effect size was represented by the weighted mean difference and its corresponding 95% confidence interval (CI). Forest plots were used to visualize the results of the meta-analysis for the available comparable variables. Imprecision was assessed using forest plot, optimal information size, and the range of CI.

### 2.13. Reporting Bias Assessment

Reporting bias was assessed using a funnel plot to visualize potential asymmetry in the distribution of study results. However, due to the limited number of comparable studies (less than 10), a statistical assessment of the funnel plot was not performed. 

### 2.14. Certainty Assessment

Assessing the certainty of evidence was conducted by evaluating the quality of evidence in five domains, namely study design, risk of bias, inconsistency, indirectness, and imprecision. Reporting bias was also considered using the Grading of Recommendations, Assessment, Development, and Evaluation (GRADE) certainty assessment of evidence [29]. The evidence was categorized into four grades, namely high, moderate, low, and very low. The GRADE tool from Cochrane demonstrated the RoB 2 through a graph providing a visual representation of the certainty in the body of evidence. All authors independently estimated the GRADE certainty and made rating adjustments by downgrading one grade (serious concern) or two grades (very serious concern) for reasons like risk of bias, inconsistency, indirectness, imprecision, and publication bias [30]. Six domains of the GRADE certainty were assessed by two authors, and any discrepancies were discussed until a consensus was reached on the final results.

## 3. Results

### 3.1. Search Results

#### Number of Retrieved Studies 

The search of the databases initially yielded 3114 literature items matching the keywords. After the removal of the duplicate abstracts by a single author (T.F.), a total of 2579 studies remained for further consideration (Figure 1).

In the initial stage, all authors collectively selected studies based on their abstracts, resulting in 95 eligible studies. Next, all authors examined the full text of all of these studies. Finally, after critical review, a total of seven eligible studies were selected for inclusion in the review [31,32,33,34,35,36,37].

### 3.2. General Information of Searched Studies

#### Quality of the Included Studies (Methodological Quality)

Results of the RoB score and a summary of the quality of the eligible studies are presented in Appendix A (Cochrane). Additionally, the RoB score summary is presented in Appendix A and the RoB score graph is shown in Appendix A, both using the GRADE approach.

Regarding domain 1, one study [37] (Sakurai et al.) was rated as having “some concerns”, while the other six studies had a “low” RoB score. For domains 2 and 4, all studies were judged as having a “low” RoB score. Concerning domain 3, two studies [32,37] (Kwon et al., Sakurai et al.) were rated as “high”, whereas the remaining five studies were judged to have a “low” RoB score. As for domain 5, one study [33] (Kawabata et al.) received a rating of “high” RoB score, while the other six studies were judged as “low”. Finally, considering the overall RoB score, three studies [32,33,37] (Kwon et al., Kawabata et al., Sakurai et al.) were categorized “high” on the RoB 2, while the other four studies were rated as “low”.

The results of the certainty assessment are presented in Table 1. The RoB 2 was categorized as either low or high. The inconsistency and indirectness of all outcomes were assessed and were found to have no serious concerns. However, due to the small sample sizes, imprecision was rated as a “serious risk” for all outcomes. The certainty of each outcome varied from very low to moderate.

### 3.3. Descriptive Results after Systematic Review

Descriptive results of this review, organized according to the PICO framework, are as follows: participant characteristics (Table 2), interventions (Table 2), comparative manners (Table 3), outcomes (Table 3), and impacts of interventions (Table 3). A summary of the efficiency categories comparing between the intervention and control groups and between the beginning and end of the intervention showed good results. Specifically, all outcome categories of each study individually demonstrated improvement, except for the “Frailty” outcome category. 

The primary descriptive results are presented in Table 2 and Table 3, with information organized systematically according to the PICO elements, providing a comprehensive understanding of the studies under review.

### 3.4. Participants

The extracted participant characteristics are presented in Table 2. Four of the included studies focused on individuals who were either prefrail or frail in terms of their frailty status [31,32,33,35].

Among the included studies, four studies identified participants as either frail or prefrail based on Fried’s criteria [38,39] or commonly used frailty criteria in Japan [38,39,40]. The other three studies [34,36,37] did not record the frailty status of participants. Additionally, three studies did not involve participants with frailty conditions.

### 3.5. Interventions (Table 2)

All seven studies [31,32,33,34,35,36,37] included in the review encompassed both DEMI and additional interventions, which can be categorized into group activities (GA), health education (HE), and community participation.

The interventions examined in our review were categorized into three main categories: “Food and diet intake”, “Exercise”, and “Other”. The following sections provide an overview of the primary outcomes observed in each category, presented in sequential order.

#### 3.5.1. “Food and Diet Intake” (Table 2)

We categorized interventions into lecture (Lc), practical exercise, GA, and consulting, instruction, and guidance. These categories were determined based on the form of intervention delivery and how these interventions were implemented, including whether they were offered by professionals to participants, conducted in a group setting, self-directed, involving Lcs or instructions, GA, home programs, or active learning (AL).

#### 3.5.2. “Exercise” (Table 2)

Types of “Exercise” interventions

Programs of “exercise” interventions were mainly categorized as follows: light exercise, muscle strengthening exercise (MS), balance exercise (BE), functional exercise (FE), functional activities (or activities of daily living [ADL]), gait exercise (GE), GA, health class, AL, and special anti-aging program (AP). One study had no access to detailed records (35). The contents of the exercise programs were as follows.

### 3.6. Special Anti-Aging Program

These exercises are widely and comprehensively recognized as being beneficial for older individuals. Their primary objectives include reducing the risk of falls and improving or preventing functional impairments. Interventions focused on fall prevention exercises were also implemented, as described in one of the studies [33].

#### “Other” (Table 2)

In addition to diet, food, and mixed interventions, other interventions also included activities such as cooking sessions, social engagement, AL, and various forms of GA.

Six studies [31,33,34,35,36,37], except one [32], incorporated not only food and exercise interventions but also added different forms of interventions. These interventions fell into the “Other” category and encompassed a diverse range of GA. Some examples of these activities included group discussions, shared experiences, community outings, information exchange, HE, AL, self-health checks, and visiting hot springs.

The programs under the “Other” category intervention were quite diverse. They extend beyond indoor settings and often involve going outdoors. Examples include group meetings centered around hobbies and interests [31], communal lunch sessions [34], and activities like HE and AL, aimed at promoting well-being.

### 3.7. Outcome (Table 3)

The variables of outcomes in each article, described in order, are shown in Table 3. We categorized those variables as follows: frailty, physical function (“Phy F”), psychological function (“Psy F”), food and dietary factors (“F”), nutritional factors (“Nu”), behavior factors like daily activities, frequency, or duration (“B”), and other factors including quality of life (QOL) (“O”). The variables for each criterion are shown in Table 3.

#### Outcomes Comparison by Meta-Analysis

The results pre- and post-interventions were synthesized by meta-analysis and are presented in Figure 2. A meta-analysis was conducted only when the number of reported outcome data from eligible studies was more than two and when data at pre- and post-intervention could be extracted. The synthesized and compared outcomes were described as follows: 

Phys F: grip strength, one leg standing (OLS), usual gait speed (UGS), maximal gait speed (MGS), and Timed Up and Go test (TUG). 

Psy F: Geriatric Depression Scale (GDS)

Food: Food Frequency Questionnaire (FFS), Diet Variety Score (DVS), and Brief Self-

Administered Diet History Questionnaire (BDSQ)

BDSQ: BDSQ fish and shellfish, BDSQ meat, BDHQ egg, BDHQ dairy products, BDHQ energy, BDHQ protein, and BDHQ Animal protein.

In Physical Function (Phy F), variables were hand grip strength, balance (OLS with eyes open, TUG test), and gait (usual and maximal gait speeds).

In Psychological Function (Psy), the variable was the GDS score.

In Food (F) and Nutrition (N), the variables were FFS, DVS, and BDHQ (meat, eggs, dairy products, energy, protein, and animal protein).

In B and O, comparable extracted data did not exist.

The results of the meta-analysis comparing variables of outcomes were as follows (Figure 2). The results of the meta-analysis showed that significant differences were considered under 0.05 and characteristic statistical data were demonstrated as MDs, 95% CIs, Z values of overall effects, and *p* values of overall effects between pre- and post-interventions.

MD in outcomes of Phys F: grip strength was NS (−1.02, CI: −2.79–0.758, Z = −1.13, *p* = 0.26), OLS was NS (MD = 0.46, CI: −8.71; 9.63, Z = 0.10, *p* = 0.92), UGS significantly improved (MD = 0.07, CI: 0.01; 0.13, Z = 2.22, *p* = 0.027), MGS was NS (MD = 0.07, CI: −0.02; 0.17, Z = 1.48, *p* = 0.139), TUG was NS (MD = −0.33, CI: −0.77; 0.11, Z = −1.48, *p* = 0.14).

MD in outcomes of Psy F: GDS was NS (MD = 0.31, CI: −0.54; 1.17, Z = 0.71, *p* = 0.477),

MD in outcomes of Food: FFS improved significantly (MD = 2.62, CI: 1.62; 3.61, Z = 5.16, *p* < 0.0001), and DVS improved significantly (MD = 35.78, CI: 33.58; 37.98, Z = 31.84, *p* < 0.001).

In the outcomes of subitems in the BDSQ: BDSQ fish and shellfish was NS (MD = 13.38, CI: −2.37; 29.14, Z = 1.67, *p* < 0.096), BDSQ meat was NS (MD = 5.34, CI: −1.24; 11.91, Z = 1.59, *p* = 0.112), BDSQ egg significantly improved (MD = 5.10, CI: 0.77; 9.43, Z = 2.31, *p* = 0.021), BDSQ dairy products significantly improved (MD = 24.09, CI: 0.54; 47.64, Z = 2.00, *p* = 0.045), BDSQ energy was NS (MD = 155.92, CI: −67.42; 379.27, Z = 1.37, *p* = 0.171), BDHQ protein significantly improved (MD = 1.56, CI: 0.30; 2.82, Z = 2.42, *p* = 0.016), and BDSQ animal protein significantly improved (MD = 2.07, CI: 0.53; 3.62, Z = 2.62, *p* = 0.009).

Reporting bias was not analyzed in detail in our review. Although we discussed visualizing the funnel plot graph for each outcome, the analysis was not adequately conducted due to insufficient information on concentration and the small size of the studies.

## 4. Discussion

In this review, we discussed the effects of DEMI to some extent, and the meta-analysis could verify the effects of limited variables, with the certainty of evidence ranging from very low to moderate. Although the narrative review in this manuscript solely demonstrated positive results in each study for the effects of DEMI on community-dwelling older adults in Japan, the results of the meta-analysis were limited. However, through the meta-analysis, significant improvements were observed in variables such as UGS, FFS, DVS, and the consumption of egg, dairy products, protein, and animal protein based on the BDHQ. According to the results of this review, we obtained some insight into implications and considerable issues. 

### 4.1. Results of Meta-Analysis

The outcomes of “Phys” slightly improved, while those of “Food” were improved in many variables.

In the outcome of “Phys”, an improvement of UGS was often reported regarding the effects of various interventions on older adults. Previous meta-analyses reported that exercise intervention improved gait speed [72], muscle strength, TUG [73], and Short Physical Performance Battery (SPPB) [74] but yielded uncertain results regarding functional performance or ADL [74]. Similarly, interventions combining nutrition and exercise were found to improve UGS; however, their impact on functional performance remained uncertain [8]. A qualitative analysis through systematic review [3] also reported uncertain results but did not include meta-analysis.

One of the reasons for the uncertain results yielded by these studies was due to the variations in participants’ frailty or functional condition and the small sample size. UGS is a representative symptom, making it easier to detect improvements. However, functional performance encompasses various physical basic functions, making it more challenging to observe significant changes in overall physical performance.

Another reason might be that the duration of interventions was too short to observe sufficient substantial effects. UGS represents a basic function for the older adults; however, improvements in functional mobilities or ADL might require a more extended period of intervention. Previous studies [75,76] reported that muscle strength improved after 12 weeks of intervention, but the duration was not sufficient to show significant changes in muscle mass. However, after 24 weeks, positive results in functional mobility were observed, prompting discussions on the potential limitation of short intervention durations in achieving significant recovery.

In the outcome of “Psy”, the improvement of the GDS score showed variable results. Hsieh et al.’s study [77], which involved home-based exercise, reported no change in depression symptoms. However, studies conducted by Singh et al. [78] and Blumenthal et al. [79] demonstrated improvement in depression syndrome among older adults was noted following home-based exercise. Many studies have reported that mental function is closely related to social functioning [80,81,82,83]. Regarding behavior or social aspects, our review could not verify the effects associated with “Psy”. Therefore, the effects of social functioning should be further researched to gain a comprehensive understanding. 

Regarding the outcome of “Food” in DEMI, FFS, DVS, and some outcomes of the BDHQ demonstrated significant improvements. The interventions related to “Food” included instructions on eating behaviors, food habits, and recommended food or ingredients, provided through instruction or counseling by nutritionists, or through information exchange and communal eating among participants. These intervention contents are related to daily activities about “Food” and are commonly practiced in clinical settings, involving aspects like increasing protein intake, diversifying food choices, and providing nutritional guidance on shopping and eating behaviors. However, the specific reasons for the improvements in the outcome of “Food” remained uncertain. Previous studies [43,84,85] have reported that nutritional status is related to social interaction. Nevertheless, we could not definitively determine whether the interventions improved dietary nutrient intake or the underlying reasons and mechanisms behind these improvements. 

### 4.2. Quality of Retrieving Eligible Studies

Previous systematic reviews on mixed nutrition and exercise interventions for the older adults have reported poor quality in some studies [3,8]. The quality of the eligible studies in our review varied from very low to moderate. Several issues remain to be addressed when designing future research studies, including concerns about blinding and the adequacy of sample sizes.

### 4.3. General Information on Interventions Extracted from the Narrative Review

Previous studies have not yet reached a conclusive determination regarding the appropriate optimal duration of intervention. Regarding exercise interventions, a previous systematic review demonstrated that durations of around 12 weeks were insufficient for achieving outcomes [73]. In our review, all seven relevant individual studies discussed showed positive outcomes. However, only two studies had a duration of more than approximately 12 weeks [35,36]. We could not definitively ascertain whether the longer durations of some studies in this review consistently yielded better outcomes. Therefore, further research to determine the appropriate duration of DEMI is desirable.

### 4.4. Characteristics of Intervention Extracted from the Narrative Review

We discussed three categories of interventions: “Food and diet intake”, “Exercise”, and “Other”.

#### 4.4.1. Food and Diet Intake

The interventions related to “Food and diet intake” were predominantly implemented through GA in our review. These GA often involved practical tasks that encompassed various stages of the eating process, for instance, cooking, shopping, preparation, and tidying up. Additionally, community participation was encouraged through these practical activities within this domain. Previous studies highlighted that eating meals alone among community-dwelling older individuals aged 65 years and older in Japan can lead to issues like obesity and malnutrition [86,87]. Therefore, practical activities related to daily living, especially those centered around food and diet, hold potential benefits for the older adult population.

Our review emphasized interventions that focused more on specific aspects of food and dietary interventions than broader nutritional considerations like supplementation. The rationale behind this emphasis is rooted in the idea that incorporating food and diet practices into daily activities is highly relevant to the wellbeing of community-dwelling older individuals.

#### 4.4.2. Exercise

We obtained similar findings to those reported in previous studies, which widely implemented similar programs.

Previous studies targeting very older adults over the age of 75 years suggested notable improvements in muscle strength [76,88]. In Grgic et al.’s meta-analysis, the muscle strength of the lower leg increased and hand grip strength was not significantly changed [88]. In Stewart et al.’s systematic review [76], three out of the four eligible studies [89] demonstrated enhancements in muscle strength through muscle and high-intensity physical training. Similarly, another systematic review and meta-analysis [72] and an RCT [90] provided evidence for the efficacy of exercises with high- or middle-to-high-load intensity exercise. Other previous studies [91,92] examined light-intensity MS exercises. Watanabe et al.’s study [91] demonstrated the efficacy of light and slow exercises using body weight for older adults aged 60–77 years [92]. Kanda et al. [93] investigated the effects of low-intensity bodyweight training with slow movements on older individuals aged 66–93 years [92]. These studies attributed the validated effects to the differentiated exercise load, which was influenced by the characteristics of the participants, especially their age. In our review, the majority of interventions focused on middle- to low-intensity training, with one exception [31]. We assumed that exercise intensity can prove effective not only at a high level but also within the middle- to low-intensity ranges, particularly for the older adult population.

Our review highlighted a unique set of exercise programs aimed at facilitating the implementation and efficacy of exercise. These programs, which formed a part of HE, FE, and ADL, were also designed to enable exercises to be carried out at home, similar to the exercises implemented in home settings [32,34].

For community-dwelling older individuals, we recommend FE and ADL interventions. These programs are tailored to participants’ daily lives, thereby promoting increased activity levels for the older adults within their own homes [93]. A study conducted in Japan demonstrated the positive impact of exercise habits adopted in middle age on the wellbeing of older adults through interviews [94]. Within the context of FE and ADL, our review encompassed BE, FE, ADL, and GE. For older adults rehabilitation, interventions that are adapted to individuals’ ADL or daily exercise habits [94] are pivotal, given the close relationship between physical functions and ADL [74,95,96]. One study examined behavior within the context of ADL [33], while another focused on FEs such as kneeling and chair stands [32]. Moreover, another study introduced GE [97], which was grounded in both exercise and nutritional interventions. The FEs within this study were adapted to each participant’s specific goals, thereby emphasizing the significance of tailoring activities to individual movements within daily life [97].

Another crucial facet of interventions within the “Exercise” domain was their incorporation of community participation, GA, HE, exercises at home, and community participation.

#### 4.4.3. Others

The majority of the eligible studies (four of seven) focused on interventions categorized as “Other” programs [32,34,36,37]. GA was used to foster engagement and motivation through various shared events. These events provided participants with opportunities to interact, discuss, and share their experiences, ultimately promoting group cohesion. Additionally, other diverse programs within this category aimed to enhance multiple aspects, such as social participation and physical activities, as outlined in Table 2.

HE programs were also implemented, including interventions like lifestyle change, AL, and communication facilitated through GA. Although the specifics of these programs were not extensively detailed in our review, they align with the broader “Other” category. The studies that incorporated these programs reported positive outcomes in each case. However, due to limitations in synthesizing the data for meta-analysis, a conclusive assessment of the efficacy of HE interventions remains challenging. 

The defining features of interventions within the “Other” category aimed to enhance behavioral changes and facilitate participant interactions. The synergistic effects of these interventions were likely derived from the interactions participants had with one another during the shared activities. The efficacy of these interactions within various activities in the “Other” interventions warrants further discussion and exploration. 

#### 4.4.4. Frailty

Four studies [31,33,35,36] that included participants who were either frail or pre-frail indicated improvements in frailty or pre-frailty status. However, the synthesis of these results does not definitively establish whether interventions can consistently lead to improvements in frailty. This uncertainty arises from the lack of comparative studies between individuals with frailty and those without. It is possible that the participants categorized as frail or pre-frail might have been present across all of the retrieved studies. This situation makes it challenging to comprehensively assess changes in frail or pre-frail participants and to determine the extent of their improvement.

#### 4.4.5. Behavior Changes

Furthermore, another critical aspect that deserves attention is the scarcity of studies focusing on the health behavior aspect of this review’s thesis, which pertains to the promotion of a healthy lifestyle among older adults. Although the number of such studies was limited, these investigations remain crucial for addressing the overall wellbeing of the older adult population.

We assumed that DEMI plays a pivotal role in enhancing the lives of the older adults. Another pivotal area for investigation pertains to behavior changes related to dietary habits. Direct measurement of outcomes about behavior changes, such as transitions through various stages of behavior change [52], was not readily apparent. However, certain outcomes indirectly demonstrated the effects of behavior changes, such as improvements in ADL, Instrumental Activities of Daily Living (IADL), QOL, and mental function.

A particularly remarkable contribution to the field was Uemura et al.’s two studies [35,36], both of which centered on participants’ health literacy, a factor that significantly influences health behavior. The AL program implemented in these studies is notably practical for integration as a preventative care initiative or as part of voluntary community health activities. Its cost-effectiveness and absence of specific equipment requirements make it easily implementable.

Throughout this review, the enhancement of the nutritional and functional status of the older adults was identified as being partly achieved through changes in their behavior. However, it is essential to note that behavior change needs prolonged and sustained treatment, often spanning at least 6 months [52]. The absence of studies that were implemented over such an extended duration is noteworthy. Additionally, interventions involving mentoring or counseling were not identified in our findings. This necessitates meticulous research into long-term studies exceeding a 6-month duration and exploring the potential benefits of mentoring or counseling interventions.

### 4.5. Novelty of This Review

Our review discusses the efficacy of DEMI for community-dwelling older adults in Japan. The novelty lies in selecting well-designed RCTs in Japan that have not been adequately discussed in previous studies. Previous studies on mixed interventions encompassing exercise and food for community-dwelling older adults in Japan were often hindered by poor designs, including single-cohort and non-RCT designs, lacking comparative analyses. Only a few studies managed to verify the efficacy, primarily through appropriate methods like meta-analysis [8]. These studies explored the impact of exercise and nutritional mixed interventions on daily eating behavior, physical activities, and healthy food habits. Some studies specifically examined the benefits of adopting a Mediterranean diet [98] or focusing on healthy food for specific purposes [42]. Studies from other countries on diet and exercise interventions have shown improvements in physical function and QOL, focusing on areas such as daily eating behavior and maintaining physical activities. However, most of these studies suffered from small sample sizes and uncertain effects. Past systematic reviews or meta-analysis [3,8] also struggled to provide clear evidence due to the scarcity of well-designed studies. 

Our review differs from previous systematic reviews on the older adults in Japan; interventions of previous studies have been designed around a combination of nutrition and exercise intervention and focused on nutritional supplementation. However, for the nutrition status of community-dwelling older adults, it is essential to consider nutrition intake behaviors in daily life. These behaviors include aspects like eating, cooking, preparing, and tidying up after meals; shopping; and relevant series of daily eating. Surprisingly, very few studies have evaluated specific areas of food and eating behaviors in this context.

In this review, we have obtained novel results, particularly regarding outcome measurements that were not commonly explored in previous studies. In studies investigating DEMI, outcomes related to behavior change were not measured in earlier research. According to Yoshimura et al.’s systematic review [8], participants with sarcopenia did not receive “food” interventions and did not engage in discussions related to behavior change.

From a quality assessment perspective, we demonstrated good quality in the included studies; however, previous studies conducted in Japan did not assess the quality of research adequately. Most reviews in Japan did not assess the quality of the literature. Only one review [8] focused on a relevant area, discussed the possibility of conducting an RCT, and assessed studies using the GRADE system. Studies included in Yoshimura et al.’s review included participants from hospitals or facilities, not solely community-dwelling older individuals [8]. Moreover, their focus was primarily on nutrition, particularly supplementation, rather than a broader exploration of food-related interventions, as in our review.

### 4.6. Limitations

This review has some limitations. First, the process of retrieving eligible studies for our review might have been subject to selection bias for several reasons. One major challenge we encountered was limiting the search to specific languages, which introduced difficulties during the selection process. This could potentially lead to literature selection bias. During the study selection process, the restriction of searching for studies both in Japanese and English might have introduced a language bias. Despite these limitations, we made efforts to minimize bias by conducting searches in both Japanese and English, as it was essential for our targeted research in Japan, i.e., focusing on community-dwelling older adults. Additionally, age was another factor that influenced participant selection and could potentially introduce selection bias. The characteristics of participants were not compared in detail, such as between different age groups. In our review, the age of participants was inclusive, with individuals aged over 65 or 70 years old. The mean age of participants ranged from 72.1 to 76.8 years. Moreover, we did not conduct a comparison between young-old and old-old age groups in all of the studies. This lack of differentiation in age groups contributes to uncertainty regarding the outcomes, such as the relationship between outcomes for the young-old and old-old populations. Issues related to the old-old age group were not researched in all studies.

Second, in each study, all participants had different statuses. One systematic review and meta-analysis [74] analyzed studies that included different functional statuses, such as health or frailty status. As a result, these studies might have estimated different functional statuses at the research baseline for their respective participants.

Third, we recognize that our selection of only seven studies may have limited the evidence level of our review due to the small number of included studies.

Finally, a critical issue in researching older adults revolves around their lifestyle, which has a significant impact on their status of ADL and IADL, food and eating habits, and exercise behaviors. These factors are unique and may vary among countries, emphasizing the importance of considering cultural differences in any study involving older adult populations. In fact, the Japanese population has Japanese-style daily living activities, food, cooking, eating style, and daily physical activities. Certainly, it is essential to recognize that differences exist among various communities in Japan, including suburban, urban, or rural areas. This review treated Japan in contradistinction to foreign countries.

The older adult population in Japan exhibits diverse lifestyles and is marked by complexity. Furthermore, previous studies have not sufficiently addressed important issues related to functional status, household situations, and the very old older adult population. In the present day, it is crucial to recognize that many older individuals, including a significant portion of the very old adults, continue to be active in the workforce. 

## 5. Conclusions

We demonstrated the efficacy of DEMI for community-dwelling older adults in Japan. Based on our findings, we recommend the implementation of practical projects that include DEMI, with a particular focus on areas that need further research. Additional research is needed to provide deeper insight. Finally, longitudinal studies are warranted, and discussions regarding the relationship of behavior change should be central to the research agenda for community-dwelling older adults in Japan. To advance the care system for older individuals in Japan, it is imperative to conduct better-designed studies that contribute to a more sophisticated and comprehensive understanding of their needs and the interventions that can best serve them.

## Figures and Tables

**Figure 1 geriatrics-09-00032-f001:**
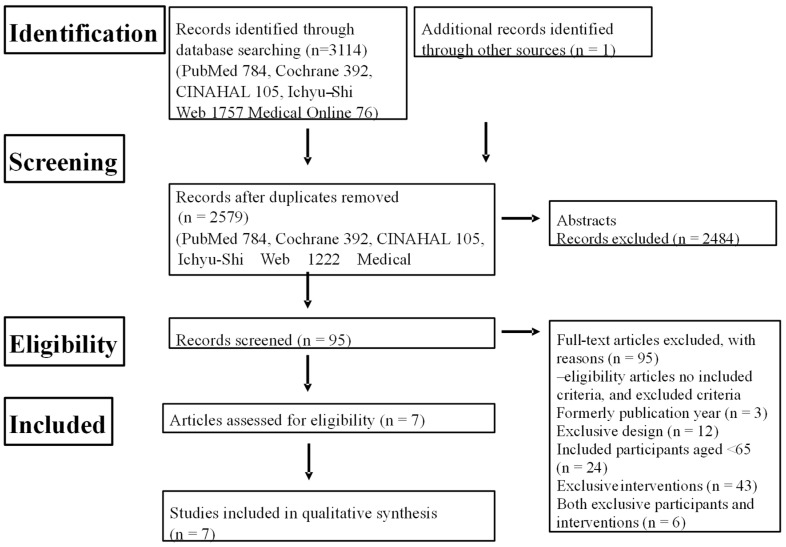
Flow chart of systematic review.

**Figure 2 geriatrics-09-00032-f002:**
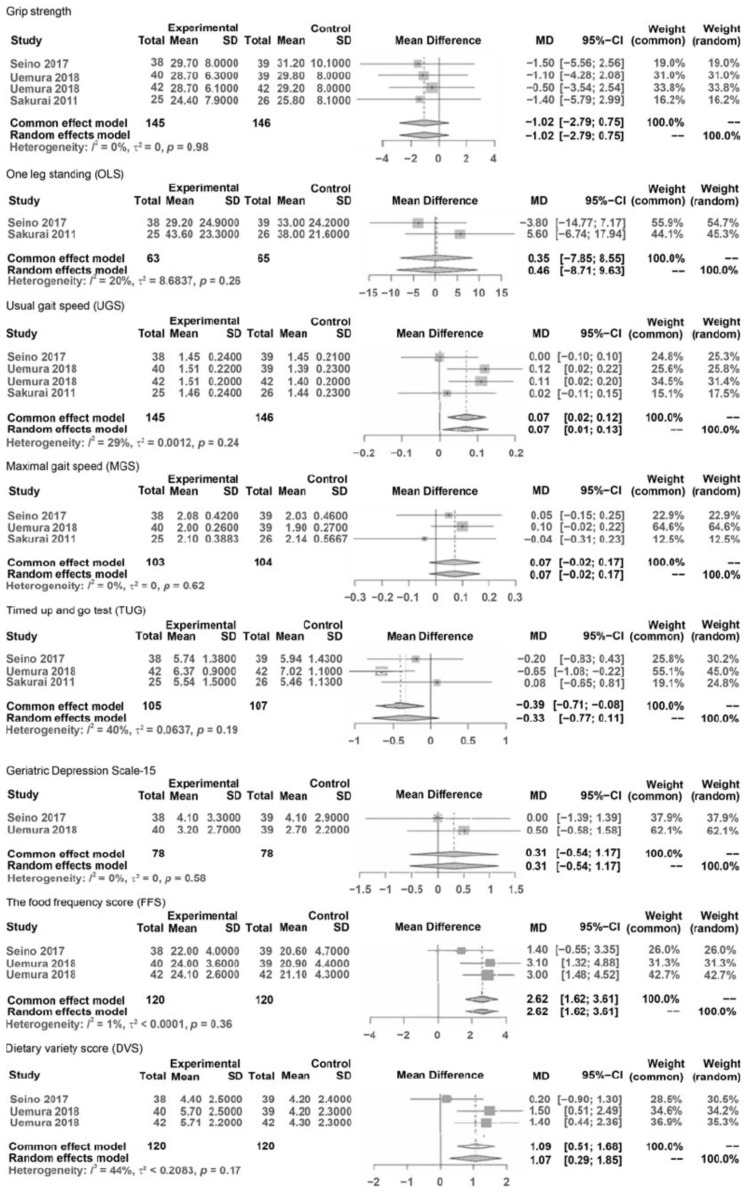
Forest plots of pooled mean differences.

**Table 1 geriatrics-09-00032-t001:** GRADE Certainty Assessment.

Certainty Assessment										
Outcome	Number of Studies	Design	RoB 2	Inconsistency	Indirectness	Imprecision	No of patients			
							Intervention	Control	MD	Certainty
Grip Strength	4	RCT (4), Seino (2017) [31], Uemura (2018) [35], Uemura (2018) [36], Sakurai (2011) [37]	Sakurai (2011) [37]: high (−2)	No serious inconsistency, I^2^ = 0%	No serious indirectness	Serious risk of imprecision as small sample size (−1)	165	167	−1.36 (95%CI −3.00; 0.29)	Very low
OLS	2	Seino (2017) [31], Sakurai (2011) [37]	Seino (2017) [31]: low, Sakurai (2011) [37]: high (−2)	No serious inconsistency, I^2^ = 0%	No serious indirectness	Serious risk of imprecision as small sample size (−1)	83	86	−0.74 (95%CI −6.42; 7.90)	Very low
UGS	4	RCT (4), Seino (2017) [31], Uemura (2018) [35], Uemura (2018) [36], Sakurai (2011) [37]	Seino (2017) [31]: low, Uemura (2018) [35]: low, Uemura (2018) [36] low, Sakurai (2011) [37]: high (−2)	No serious inconsistency, I^2^ = 29%	No serious indirectness	Serious risk of imprecision as small sample size (−1)	145	146	0.07 (95%CI −0.01; 0.13)	Very low
MGS	3	RCT (4), Seino (2017) [31], Uemura (2018) [35], Sakurai (2011) [37]	Sakurai (2011) [37]: high (−2)	No serious inconsistency, I^2^ = 0%	No serious indirectness	Serious risk of imprecision as small sample size (−1)	123	125	−0.07 (95%CI −0.02; 0.16)	Very low
TUG	3	RCT (4), Seino (2017) [31], Uemura (2018) [35], Sakurai (2011) [37]	Sakurai: high (−2)	No serious inconsistency, I^2^ = 40%	No serious indirectness	Serious risk of imprecision as small sample size (−1)	125	128	−0.14 (95%CI −0.62; 0.33)	Very low
GDS	2	RCT (4), Seino (2017) [31], Uemura (2018) [35]	No serious risk of bias, low	No serious inconsistency, I^2^ 0%	No serious indirectness	Serious risk of imprecision as small sample size (−1)	78	78	−0.31 (95%CI −0.54; 1.17)	Moderate
FFS	3	RCT (4), Seino (2017) [31], Uemura (2018) [35], Uemura (2018) [36]	No serious risk of bias, low	No serious inconsistency, I^2^ 1%	No serious indirectness	Serious risk of imprecision as small sample size (−1)	120	120	2.62 (95%CI 1.62; 3.61)	Moderate
DVS	3	RCT (4), Seino (2017) [31], Uemura (2018) [35], Uemura (2018) [36]	No serious risk of bias, low	No serious inconsistency, I^2^ 44%	No serious indirectness	Serious risk of imprecision as small sample size (−1)	120	120	35.78 (95%CI 33.58; 37.98)	Moderate
BDHQ Fish and shellfish	2	RCT (4), Seino (2017) [31], Kawabata (2015) [33]	Seino (2017) [31]: low, Kawabata (2015) [33]: high (−2)	No serious inconsistency, I^2^ 58%	No serious indirectness	Serious risk of imprecision as small sample size (−1)	59	61	13.38 (95%CI −2.37; 29.14)	Very low
BDHQ Meat	2	RCT (4), Seino (2017) [31], Kawabata (2015) [33]	Seino (2017) [31]: low, Kawabata (2015) [33]: high (−2)	No serious inconsistency, I^2^ 0%	No serious indirectness	Serious risk of imprecision as small sample size (−0.5)	59	61	5.34 (95%CI −1.24; 11.91)	Very low
BDHQ Egg	2	RCT (4), Seino (2017) [31], Kawabata (2015) [33]	Seino (2017) [31]: low, Kawabata (2015) [33]: high (−2)	No serious inconsistency, I^2^ 0%	No serious indirectness	Serious risk of imprecision as small sample size (−1)	59	61	5.10 (95%CI 0.77; 9.43)	Very low
BDHQ Dairy products	2	RCT (4), Seino (2017) [31], Kawabata (2015) [33]	Seino (2017) [31]: low, Kawabata (2015) [33]: high (−2)	No serious inconsistency, I^2^ 19%	No serious indirectness	Serious risk of imprecision as small sample size (−0.5)	59	61	24.09 (95%CI 0.54; 47.64)	Very low
BDHQ Energy	2	RCT (4), Seino (2017) [31], Kawabata (2015) [33]	Seino (2017) [31]: low, Kawabata (2015) [33]: high (−2)	No serious inconsistency, I^2^ 0%	No serious indirectness	Serious risk of imprecision as small sample size (−1)	59	61	155.92 (95%CI −67.42; 379.27)	Very low
BDHQ Protein	2	RCT (4), Seino (2017) [31], Kawabata (2015) [33]	Seino (2017) [31]: low, Kawabata (2015) [33]: high (−2)	No serious inconsistency, I^2^ 31%	No serious indirectness	Serious risk of imprecision as small sample size (−1)	59	61	1.56 (95%CI 0.30; 2.82)	Very low
BDHQ Animal protein	2	RCT (4), Seino (2017) [31], Kawabata (2015) [33]	Seino (2017) [31]: low, Kawabata (2015) [33]: high (−2)	No serious inconsistency, I^2^ 44%	No serious indirectness	Serious risk of imprecision as small sample size (−1)	59	61	2.07 (95%CI 0.53; 3.62)	Very low

BDHQ: Brief Self-Administered Diet History Questionnaire, RCT: randomized controlled trial, GDS: Geriatric Depression Scale, FFS: Food Frequency Questionnaire, DVS: Diet Variety Score, UGS: usual gait speed, OLS: one leg standing test, MGS: maximal gait speed, TUG: Timed Up and Go test; GRADE: Grading of Recommendations, Assessment, Development, and Evaluation, MD: mean difference, CI: confidence interval.

**Table 2 geriatrics-09-00032-t002:** Participants (Characteristics of Participants Included in Study) and Intervention.

	Participants							Intervention					
First author, year *	Recruitment method **	Subjects for analysis, sample size, and mean age of participants, % Women ***	Health (comorbidity)/functional status	Frailty and sarcopenia definition used in study	Level of frailty, prefrailty, and sarcopenia of participants pre-intervention	Inclusion criteria	Exclusion criteria	Duration, frequency, time	Participated profession	Food and diet intake intervention ^††^	Exercise intervention (professional, program, exercise load) ^†††^	Other ^‡^	Intervention of Control and Placebo groups
Seino (2017) [31]	The Hatoyama Cohort Study, public offering, brochure, mail	N; 77, Intervention Group (Int) 1: n; 38, 74.9 ± 5.3, 14 (36.8%), Intervention Group (Int) 2: n; 39, 74.3 ± 5.6, 10 (25.6)	Comorbidity exists, but mild symptoms and functional independently	CL15 [38,39,40]	Prefrail or frail, a score of 2 or higher (CL-15)	Age ≥ 65, participants in the Hatoyama Cohort Study and a score of 2 or higher on the CL-15	Routine participation in health promotion activities, presence of a serious or unstable illness	Duration of intervention and FU; 3 months, FU; PI to FU: 3 months, 2/w, 100 min/session, resistance exercise: 60 min, rest 10 min, nutritional or psychosocial program: 30 min each, 3-month period	NR	Lc, GA (checklist, using a map to find restaurants and supermarkets)	MS (toe and heel raises, knee lifts, knee extension, rowing with a resistance band while seated. Lateral leg raises and standard squats. The repetitions and sets increased progressively. Two sets of 20 repetitions for each exercise in the final month. Intensity: self-rated, perceived exertion of “somewhat hard”)	GA (The psychosocial program, discuss (hobbies, interests, experiences (neighborhood and community environment), participation)	Because of COT design, no C
Kwon (2015) [32]	Public offering, mail, brochure, town bulletin	N; 79, 76.8 (70–84), 100%, participants, completed the 12-week, exercise and nutrition intervention group (EN): n; 26, 76.5 ± 3.8, 100%, exercise only group (E), n; 25, 77.0 ± 4.2, control (C): n; 28, 76.9 ± 3.9, 100%	Comorbidity exists, but mild symptoms and functional independently	Freid’s criteria [41,42]	Prefrail or frail	Age ≥ 70, muscle weakness (handgrip strength ≥ 23 kg or slow gait speed ≤ 1.52 m/seconds (lowest quartile of timed usual walking speed at baseline).	Serious musculoskeletal diseases, serum albumin ≥ 4.5 mg/dL, serious musculoskeletal conditions, taking calcium or vitamin D supplements.	Duration of intervention and FU; BL to PI: 3-month, PI to FU: 6 months, EN, E; exercise intervention program: 12 w, 1/w, 1 h, nutritional intervention program: 12 w, 1/w, 2–3 h	Certified dietician, fitness trainer, 1 physician and 2 assistants.	Lc, GA and PE (cooking class, nutrition guidance, eating together, preparation, washing dishes, and tidying up)	GA, MS, BE, using body weight and theraband, dumbbells, and balls, four class, given diagrams and explanations, done at home, checklist, walking, kneeling, chair stands, individually tailored		HE general health education session (1/month for a total of 3 sessions during the 12-week intervention period), physician, certified health fitness trainer, and dietician provided the participants with information on physical training for preventing falls and urinary incontinence as well as a dietary guideline for healthy aging. After all int, 12 w, exercise and nutritional program as in the same manner for EN and E.
Kawabata (2015) [33]	Mailed to subjects of the Hatoyama Cohort Study, public offering, recruitment manners were NR	N; 47, Int: n; 21, 75.7 ± 5.4, 6(28.6%), C: n; 22, 74.7 ± 5.4, 4 (18.2%)	NR	CL15	Prefrail or frail	Age ≥ 65, CL-15 ≥ 2 points	CL-15 ≤ 1 point (non-frail), severely disability	Duration of intervention and FU; BL to PI: 3 months, PI to FU: 3 months, exercise: 2/w, 60 min/session, 20 times, nutrition: 1/w, 30 min/session, 10 times, social participation: 1/w, 30 min/session, 10 times	Certified dietician	GA, CIG, PE (instruction of food and nutrition, checking up on food, cooking, shopping, information exchange, eating together, environment of food, making map)	GA, MS, body weight, tube, BE, GE, ADL, Behavior in ADL, fall preventing exercise, promote going outside, making restaurants and shopping map	Social (community) participation, checking up on home, neighborhood, giving information for health center, health activities in the community	After intervention for Int: service program (same program as Int)
Takai (2013) [34]	Recruitment from community neighborhood association	N; 44, Int: n; 23, 72.4 ± 3.9, 11 (47.8%), C: n; 21, 69.6 ± 5.8, 9 (42.9%)	NR	NR	NR	NR	NR	Duration of intervention and FU; BL to PI: 2 months, PI to FU: 1 month, frequency about 1/w ^†^, 2 months (intervention), 1 month (follow up), 5 times (group exercise 5 times, nutrition classes 3 times, exercise 2 times)	Certified dietician, physical therapist	Lc, PE, GA (Lc about food health and nutrition, eating together with stuff, preventive care program called Take 10 class by Japanese ministry of health)	GA, group exercise, STR, MS, GW, Checking up self-exercise, home exercise	GA (eating together lunch meal)	Only lunch (meal) distribution, no class, after intervention for Int: GA (light exercise, gymnastics)
Uemura (2018) [35]	Town bulletin	N; 79, Int: n; 40, 72.1 ± 4.5 (65–83), 70.0%, C: n; 39, 71.5 ± 4.4 (65–85), 69.2%	Comorbidity exists, but mild symptoms and functional independently	Diagnosis of all participants 64/79, intervention group 31/40 were prefrail, but definition or criterion of prefrail were NR	Prefrail or healthy	Age ≥ 65	Prefrail or healthy, no certification for LTC, ADL independent, ≤ 24 Mini Mental State Examination (MMSE), no apparently cognitive disorders, no restriction of exercise by circulatory, pulmonary, neurologically, or orthopedic disease	24 w, 1/w, 90 min	NR	HE, AL (information about food and nutrition to improve low nutrition, muscle strength	HE, AL (walking, aerobic exercise, MS, recreation)	HE, AL (health information collection, making health and walking map, looking back elderly health and behavior)	No intervention
Uemura (2018) [36]	Advertisements in localgovernment public relations magazines	N; 84, Int: n; 42, 72.1 ± 4.6, 69.1%, C: n; 42, 71.6 ± 4.4, 71.4%	Comorbidity exists, but mild symptoms and functional independently	NR	NR	Age ≥ 65	Certified for LTC, non-independent ADL, ≤23 Mini Mental State Examination (MMSE), general cognitive disorders, restriction of exercise by circulatory, pulmonary, or orthopedic disease	24 w, 1/w, 90 min	Physical therapist, physical education committee members	HE, AL (information about food and nutrition to improve low-nutrition, muscle strength	HE, AL (walking, aerobic exercise, MS, recreation)		No intervention
Sakurai (2011) [37]	Public offering, recruitment manners were NR	N; 60 (72.7 ± 6.0, 65~93), In: n: 31, 73.2 ± 6.9, 71.0%, C: n; 29, 73.0 ± 4.9, 69.0%	Medical check by physician, no onset of stroke or circulatory organ accidents	NR	None	Age ≥ 65, community-dwelling	Restriction for participation in exercise class by physician	Duration of Intervention: Int 1: 3 months. Duration of FU: only Int 1, PI to FU: 3 months, 2/w, 90 min/session	Registered dietician, public health nurse, licensed exercise coach,	6 times, Lc, PE, GA) (weight control, nutrition, cooking, information exchange, health behavior change, checking-up)	GA (exercise class, 11 times, MS (body weight or elastic tube), tailored (position, program), intensity: self-rated, perceived exertion of “somewhat hard”“ (rate of perceived exertion 12–14: somewhat hard)	Hot spring, 30 min	1/month, Lc, health class unrelated contents to intervention of Int

* First author and publication year; ** Name of project, recruitment method; *** N is the number of all subjects, mean age, SD, n is the number in the intervention group or control group, mean age (SD), %: % of women; n = number receiving intervention, mean age (SD), % women, Int: intervention group, C: control group, E: exercise group. ^†^ We decided on the frequency from times/duration. ^††^ Food and diet intake intervention, GA: group activities, Lc: lecture, PE: practical exercise. ^†††^: Exercise intervention (professional, program, exercise load), HE: health education, MS: muscle strengthening exercise. ^‡^ HE: health education, specific activities including Lc, GA, instruction, giving information, participation, group activities, discussion, health education, fall prevention, checking home environment, going to museums, and community events; AL: active learning, BL: baseline, CL-15: Check-List 15, COT: crossover trial, FU: follow up, Kihon-Check-List, MMSE: Mini Mental State Examination, NR: not recorded, PI: post-intervention, SD: standard deviation.

**Table 3 geriatrics-09-00032-t003:** Comparison, Outcome, and Impact (Measures of Frailty and Outcomes of Study).

			Impact of Intervention on Frailty Outcome	Impact Change with Time		Impact Difference between Groups
First author, year *	Comparison design between groups (groups that were compared and timing) **	Outcome *** (Frailty, Phy F, Psy F, Nu, F, B, O) ^†^		Change in impact from BL to PI ^††^	Change in impact from PI to FU	Comparison between each group
Seino (2017) [31]	COT (Intervention Group 1 (Int 1) vs. Intervention Group 2 (Int 2), Int 1 (PI vs. FU)	The primary outcome: CL-15 [38,39,40], frailty status. The secondary outcomes: Phy F: hand grip strength (HGS), OLS with eyes open, usual and maximum gait speeds (GS), TUG test, Nu: body Mass Index, blood sample, F: food intake and dietary variety using a self-administered questionnaire [43], Food Frequency Score (FFS) [44], Dietary Variety Score (DVS) [43], brief-type self-administered diet history questionnaire (BDHQ) [45,46], Psy F: the Japanese version of the Short-Form Health Survey to assess health-related quality of life [47], the 15-item Geriatric Depression Scale short-form (GDS) [48], original “self-rated health questionnaire”, original “neighborhood attachment questionnaire”, O: Check List for Vivid Social Activities [49]	CL15 (Int 1 (BL to PI, B to FU), Int 2 (B to FU))	Int 1 of COT: yes, TUG, GDS, Weight, BMI, DVS, FFS, Protein, Animal protein, Vitamin B6, Vitamin B12, Folic acid, Calcium, Iron, Zinc. Int 2 of COT: yes, HGS, OLS, social participation and voluntary activity, Weight, BMI, Total cholesterol, Hemoglobin, Zinc, Meat	Yes: FU (only Gr 1), (TUG, GDS, Weight, BMI, DVS, FFS, Vitamin B6). Gr 2: No FU	Yes (interaction)
Kwon (2015) [32]	Exercise and nutrition group (EN), exercise only group (E), control; group (C), BL vs. PI vs. FU	Physical performance, HRQOL Phy F: handgrip strength, balance (OLS with eyes open), GS (usual walking speed). Skeletal MM: bioelectrical impedance analysis (INBODY 3.2; Biospace Co., Ltd., Seoul, Korea, using eight electrodes, two on each hand and foot), F: dietary variety score (DVS) [50]), O: HRQOL: SF-36 [51]		B vs. PI: EN: none, HRQOL (role physical, bodily pain, role emotional, physical component summary), E: HG, HRQOL (Mental health)	Post vs. FU: HS: declined significantly by follow-up. Gr 1: HG, HRQOL (physical functioning, role physical, bodily pain, vitality, role emotional, physical component summary, mental component summary). Gr 2: HRQOL (bodily pain,). Gr 3: HRQOL (bodily pain)	Int (EN) vs. Int (E) vs. C: NR
Kawabata (2015) [33]	Int vs. C, BL vs. PI vs. FU	Main outcome: frailty (CL-5 [38]). Sub outcomes: Nu: blood sampling, F: status of feeding (BDHQ [45,46], dietary variety score (DVS) [43])	Frail score (frailty check list, BL to PI, BL to FU)	B vs. PI: yes, CL-15, score of food intake, food intake (protein, animal protein, animal protein ratio), food (fish, egg)	B vs. FU: yes, Int: home boundness, BMI, protein intake	Int vs. C: yes, CL-15 (home boundness), nutrient intake (energy ratio (protein, animal protein), animal protein ratio), food intake (food, fish, egg)
Takai (2013) [34]	Int vs. C, BL vs. PI vs. FU	Questionnaire about frequency of going out, experience of falls, frequency of exercise, self-efficacy about exercise [52], behavior change about exercise (Prochaska’s TTM [53]), subjective feeling of health, chair stand test (CST) (5 times) [54], TUG		B vs. PI: yes, Int Gr: frequency, duration, self-perceived health efficacy	B vs. FU: yes, Int Gr: frequency	Int Gr vs. C: post: yes, frequency, time, self-efficacy for exercise; follow up: frequency, self-efficacy for exercise
Uemura (2018) [35]	Int vs. C, BL vs. PI	Main outcome: Psy F (apathy scale [55], GDS-15 [56]), Phy F (5 m ordinary gait speed, 5 m max gait speed, 5 times chair stand (CS5) [57], grip strength), lifestyle (amounts of physical activity [58], eating behavior, self-efficacy), F (dietary variety score [59]), B (self-efficacy for health promotion scale [60]), frailty (pre-frailty status based on Freid’s frailty phenotype [61], frailty based on Freid phenotype [61]	Int: the rate of pre-frailty significantly decreased (BL to PI)	B vs. PI: yes, Int: Apathy Scale, gait speed, chair stand test, steps/day, Mets*hour/week, the dietary variety score, self-efficacy for health promotion scale, number of prefrail		(Interaction) Intervention Group vs. Control Group: yes, Apathy Scale, chair stand test, steps/day, the dietary variety scale, self-efficacy
Uemura (2018) [36]	Int vs. C, BL vs. PI	Main outcome: Health Literacy Scale-14 (HLS-14) [62], 16-item European Health Literacy Survey Questionnaire (HLS-EU-Q16) [63,64,65]. Sub outcomes: Cognitive function: processing speed, digit symbol coding subset of Wechsler Adult Intelligence Scale-Ⅲ (WAIS-Ⅲ) [66]; verbal fluency [67]; working memory, digit span test [66]; memory, Scenery Picture Memory Test (SPMT) [68]. Phy F: amounts of physical activity Phy F: hand grip; 5 m gait speed (ordinary); balance, TUG; physical activity (amounts of physical activity [58]. F: The Food Frequency Score (FFS) [44]		B vs. PI: yes		Intervention Group vs. Control Group: (two-factor interaction), yes (HLS-14, the disease prevention domain of the HLS-EU-Q16, category verbal fluency test, scenery Picture Memory Test, and Timed Up and Go test scores; gait speed; number of steps per day; physical activity levels; and Dietary Variety Scores)
Sakurai (2011) [37]	RCT, COT, Int vs. C, then COT	Self-efficacy for exercise, O: HRQOL (SF-8) [69], Psy F: WHO-5 [70], scale of psychological independence [71], Phy F: grip strength, OLS with eyes opened, 5 m ordinary and maximal gait speed, TUG		Yes (Int: [B to PI]), after 3 months: GS, OLS with eyes-open. C: GS, OLS with eyes open, the World Health Organization Well-Being Index (WHO-5) scores)	Int Gr: 3 months from int end, GP, OLS WHO-5 (well-being index)	Yes (interaction): GP, OLS

* First author and publication year, ** Int: intervention group, C: control group, EN: exercise plus nutrition group, E: exercise group, BL: baseline, PI: post-intervention, FU: follow up, *** BDHQ: brief-type self-administered diet history questionnaire, CL-15: Check-List 15, HGS: hand grip strength, HLS-14: Health Literacy Scale-14, HLS-EU-Q16: 16-item European Health Literacy survey Questionnaire, Kihon-Check-List, OLS: one leg standing, TUG: Timed Up and Go test, MM: muscle mass, ^†^ Frailty: Freid’s phenotype, CL-15, Phy F: physical functions, Psy F: psychosocial functions, Nu: nutritional status, F: food intake, B: behavior (frequency, duration), ^††^ yes: significantly improved, B: baseline; COT: crossover trial.

## Data Availability

The data presented in this study are available within the article.

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
