# Peer review of "Diet, Food Intake, and Exercise Mixed Interventions (DEMI) in the Enhancement of Wellbeing among Community-Dwelling Older Adults in Japan: Systematic Review and Meta-Analysis of Randomized Controlled Trials"

_geriatrics, 2024, doi:10.3390/geriatrics9020032_

Round 1
Reviewer 1 Report
Comments and Suggestions for Authors
The present review is an interesting paper that discusses the effectiveness of DEMI (Diet and food intake – exercise – mixed interventions) for community-dwelling older adults in Japan. Well-designed RCTs in Japan that have not been adequately discussed in previous studies were selected.
Some novel results were explored, for example, outcomes related to behavior change that were not measured in earlier research.
The authors demonstrated the effectiveness of DEMI for community-dwelling older adults in Japan.
I have no specific comments because the paper seems to be of good quality.
Reviewer 2 Report
Comments and Suggestions for Authors
I would appreciate being a reviewer for your valuable submission to Geriatrics. I have some ideas and questions for your manuscript. I hope those might be helpful to improve your study.
1. Materials and Methods part is confusing and unclear to the reader. You should rewrite it specifically for understandability for any readers. Ex) Which languages do you focus on or reject? In part 2.10., primary and secondary outcomes were hard to understand. Synthesis methods are also unclear.
2. As a meta-analysis research, seven kinds of literature used to synthesize were relatively small. Because of this limitation, all of your research results have a severe limitation as evidence.
3. You use so much literature to write this manuscript. It is suitable for you to rewrite it as a literature review.
Reviewer 3 Report
Comments and Suggestions for Authors
The article titled "Diet, Food Intake, and Exercise Mixed Intervention (DEMI) in the Enhancement of Wellbeing among Community-Dwelling Older Adults in Japan: Systematic Review and Meta-Analysis of Randomized Controlled Trials" aims to assess the effectiveness of DEMI for community-dwelling older adults in Japan through a systematic review with meta-analyses. It addresses a relevant topic and generally employs appropriate and consistent methods. However, some aspects require revision and/or clarification to enhance the manuscript's quality.
-The introduction of the article lacks a clear logical progression and fails to explicitly highlight the review's differentiators. A clearer structure and emphasis on the review's unique aspects are needed in both the introduction and discussion sections.
-Ensure strict adherence to the PRISMA guidelines, including the recommended PRISMA flowchart.
-Provide further explanation on the exclusion criteria regarding severe disability. I could not understand it.
-The GRADE level of evidence is crucial information that must be included in the abstract. Furthermore, (in results section) present certainty assessment after the meta-analyses.
-Clarify how mean differences were calculated in studies with both pre- and post-values for each group.
Confirm whether the subtraction of before-and-after values within groups followed by subtraction between groups was conducted. Also, provide or explain the correlation coefficient value used in the synthesis.
If studies with both pre- and post-values do not exist, point out this limitation of the literature.
Minor comments:
-Replace "effectiveness" with "efficacy" to accurately reflect the study's evaluation.
-Insert a source reference on line 45.
-Clarify the statement regarding the lack of significance in certain studies (line 65).
-Add direction indicators for beneficial associations in Figure 2.
-Revise the formatting of tables presenting descriptive data for clarity.
Comments on the Quality of English LanguageIt is advisable to enhance the English writing style in certain sections and improve the logical organization of writing, particularly in the introduction and results sections.
Round 2
Reviewer 2 Report
Comments and Suggestions for Authors
I appreciate your work as a reviewer again for your valuable manuscript.
Your manuscript is fully improved and has scientific significance.
Thank you and warm regards.